# An *in vitro* study to assess the effect of hyaluronan-based gels on muscle-derived cells: Highlighting a new perspective in regenerative medicine

Antonietta Stellavato[1]*, Lucrezia Abate[1], Valentina Vassallo[1], Maria Donniacuo[2], Barbara Rinaldi[2‡], Chiara Schiraldi[1‡]*

1 Department of Experimental Medicine, School of Medicine, University of Campania "Luigi Vanvitelli," Via L. De Crecchio, Naples, Italy, 2 Unit of Pharmacology, Department of Experimental Medicine, University of Campania "Luigi Vanvitelli", Naples, Italy

‡ Barbara Rinaldi (BR) and Chiara Schiraldi (CS) share last authorship.
* chiara.schiraldi@unicampania.it (CS); antonietta.stellavato@unicampania.it (AS)

**Data Availability Statement:** All relevant data are within the manuscript and its Supporting Information files.

## Abstract

Hyaluronan (HA) is a nonsulfated glycosaminoglycan that has been widely used for biomedical applications. Here, we have analyzed the effect of HA on the rescue of primary cells under stress as well as its potential to recover muscle atrophy and validated the developed model *in vitro* using primary muscle cells derived from rats. The potentials of different HAs were elucidated through comparative analyses using pharmaceutical grade a) high (HHA) and b) low molecular weight (LHA) hyaluronans, c) hybrid cooperative complexes (HCC) of HA in three experimental set-ups. The cells were characterized based on the expression of myogenin, a muscle-specific biomarker, and the proliferation was analyzed using Time-Lapse Video Microscopy (TLVM). Cell viability in response to $H_2O_2$ challenge was evaluated by 3-[4,5-dimethylthiazole-2-yl]-2,5-diphenyltetrazolium bromide (MTT) assay, and the expression of the superoxide dismutase enzyme (SOD-2) was assessed by western blotting. Additionally, in order to establish an *in vitro* model of atrophy, muscle cells were treated with tumor necrosis factor-alpha (TNF-α), along with hyaluronans. The expression of Atrogin, MuRF-1, nuclear factor kappa-light-chain-enhancer of activated B-cells (NF-kB), and Forkhead-box-(Fox)-O-3 (FoxO3a) was evaluated by western blotting to elucidate the molecular mechanism of atrophy. The results showed that HCC and HHA increased cell proliferation by 1.15 and 2.3 folds in comparison to un-treated cells (control), respectively. Moreover, both pre- and post-treatments of HAs restored the cell viability, and the SOD-2 expression was found to be reduced by 1.5 fold in HA-treated cells as compared to the stressed condition. Specifically in atrophic stressed cells, HCC revealed a noteworthy beneficial effect on the myogenic biomarkers indicating that it could be used as a promising platform for tissue regeneration with specific attention to muscle cell protection against stressful agents.

**Funding:** National grant MIUR PON03PE_00060_3 "Sviluppo e sperimentazione di molecole ad azione nutraceutica e cosmeceutica" MIUR Ministero dell'Universita' e della Ricerca Scientifica. Regional Competence Center in Industrial Biotechnologies (Bioteknet) S.C.P.A funded short term grants for A. S. and L.A.

**Competing interests:** The authors have declared that no competing interests exists.

## Introduction

Diverse physiological and pathological conditions such as inactivity, aging (i.e., age-related sarcopenia), starvation, diabetes, cachexia, and cancer can cause reduced synthesis and increased breakdown of muscle proteins, leading to lessened muscle mass, known as muscle atrophy [1, 2]. Skeletal muscle atrophy is an important clinical disorder mediated by the activation of proteolytic systems inducing muscle weakness and mass reduction [3]. At the molecular level, the atrophy is associated with impaired protein metabolism in several physiological and pathophysiological conditions [4, 5]. Furthermore, the maintenance of skeletal muscle mass is based on a balance between the synthesis and degradation of muscle regulatory proteins. Specifically, atrophy resulted from an increase in protein degradation, loss of muscle mass [6, 1], and a reduction of protein synthesis (Fig 1). This process is predominantly regulated by myogenic transcription factors, the atrogenes, including FoxO3a (Forkhead box (Fox)-O 3), atrogin, also known as MAFbx1, muscle-specific ring finger protein (MuRF-1) [6], and myogenic regulatory protein such as myogenin and desmin [7], and these factors are used as biomarkers of muscle functionality [8]. Reactive oxygen species (ROS) production represents one of the most prominent events during the contractile muscle activity, suggesting that it could influence muscle-specific function. It has also been shown that ROS accumulation promoted the activation of proteolytic systems, leading to atrophy, and the degradation of muscle tissue [9]. However, the specific molecular mechanisms underlying the cell damage have not been fully

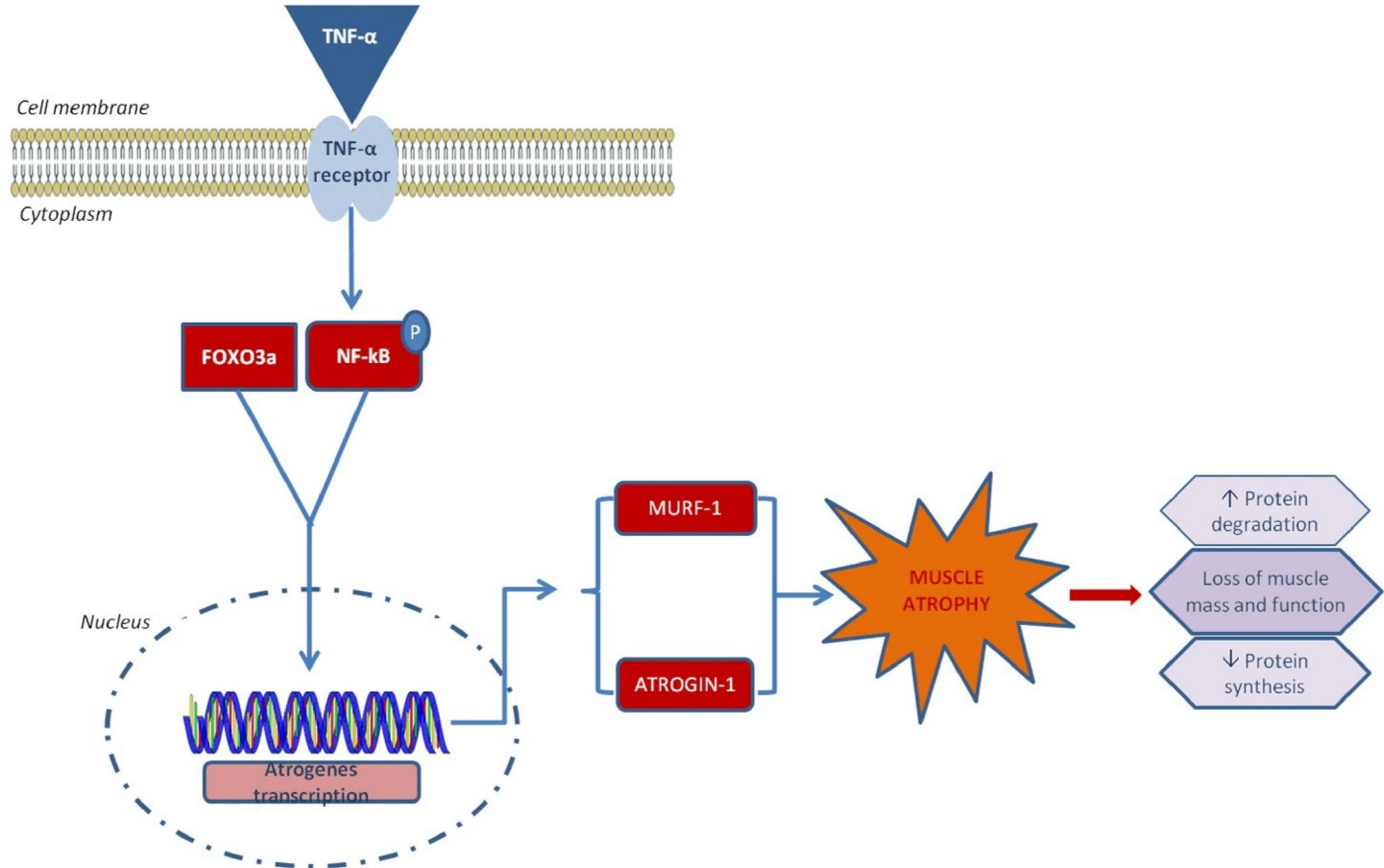

**Fig 1. Schematic description of the *in vitro* atrophy model and related signaling pathway investigated.**

explored. Several studies [10, 11] have highlighted the role of oxidative stress in atrophic muscle resulting from an imbalance between the cellular antioxidant systems and ROS production. High levels of ROS redox status and weakened antioxidant defense system are among the major contributing factors toward atrophy [12], thereby requiring a medium that could inhibit or counteract the biochemical pathways involved in cellular stress and damage.

With an objective to explore the molecular mechanisms underlying cellular damage and development of a model to recover the cells from stressful conditions, we have analyzed the potential of hyaluronan (HA), the sole natural non-sulfated glycosaminoglycan (GAG), which is ubiquitously expressed in the extracellular matrix (ECM) of mammals [13]. HA is a hygroscopic molecule that is able to structurally organize the ECM by complexing with other ECM macromolecules. Due to its rheological and biochemical properties, HA has been used as an active component in a broad range of class III medical products [14,15]. The fact that linear HA with different molecular weights produces different effects is well documented, and currently, many formulations based on linear and/or chemically cross-linked HA are used in dermo-aesthetic, wound healing, and ophthalmic applications [16]. Additionally, as a result of its natural presence in the synovial fluid, joint capsule, and articular cartilage, HA is widely used in the treatment of osteoarthritis or rheumatoid arthritis [17–19]. In addition to linear HAs, the novel stabilized hybrid cooperative complexes (HCC) derived from high and low molecular weight HA through NAHYCO$^{TM}$ technology has been reported to be used in several *in vitro* studies based on different cellular models [20]. HCC can be defined as "physical gels," in which the interactions between long and short HA chains are optimized without changing the structure of disaccharide units and without introducing other exogenous "chemical compounds." As previously reported, this formulation protects the high molecular weight hyaluronan (HHA) from enzymatic degradation, and this fact is expected to confer the longer persistence to the product *in vivo* [21, 22]. It can be expected that the typical rheological properties of HHA, as well as the biological action (e.g., receptor interaction/biochemical cascade), are more persistent in this new preparation than that in linear HA formulations. Besides, the biological properties of HCC have already been tested on several cell cultures used as representative models for tissue regeneration [22–24]. Recently, HCC has been proved to be effective on cell co-culture models subjected to stressful conditions to mimic injuries [25]. Furthermore, it is known that HA is the subject of the ROS attack, which leads to changes in its structure modulating oxidative condition [26–28]. In the present study, we have explored the comparative potentials of linear and hybrid HAs through three different experimental set-ups with respect to increasing cell growth and proliferation and rescuing cells under oxidative stress. Moreover, we have developed an *in vitro* model of muscle atrophy using rat muscle-derived cells insulated with TNF-α to mimic atrophy [29]. in order to identify the mechanism underlying muscle atrophy and to assess the potential effect of HAs on the recovery of muscle atrophy. The model developed here could be helpful in exploring the specific function of HAs not only as antioxidants [28], but also as molecules capable of modifying the ECM structure with potential in tissue remodeling and engineering.

## Materials and methods

### Materials

High molecular weight (HHA; MW 1400±200 kDa) and low molecular weight (LHA; MW 100 ±20 kDa) HA were provided by Altergon (Altergon Italia Srl, Italy). These are extensively purified fermentative HA derived from *Streptococcus equi* ssp. *equi*, at pharmaceutical grade (e.g. purity >95%, water content <10%, EU/mg <0.05, and very low metal content). Hybrid cooperative complexes of hyaluronic acid (HCC), obtained through a patented process [20] named

NAHYCO™ technology, were provided in the form of sterile syringes from IBSA Italia (IBSA Farmaceutici Italia Srl, Italy). A final concentration of 1.6 mg/mL was used in these experiments.

## Methods

**Rat muscle-derived cell isolation and experimental set-up.** Skeletal muscle tissue was obtained from rat quadriceps (*Mus musculus*; Rodentia, Muridae; Envigo, Milano, Italy, weighing 200–250 g), under anesthesia with pentobarbital (50 mg/kg, i.p.). The experimental procedures were carried out during the light phase and were approved by the Animal Ethics Committee of the University of Campania "L. Vanvitelli" (Naples, Italy). Animal care was in compliance with Italian (D.L. 116/92) European Economic Community (EEC) (O.J. of E.C. L358/1 18/12/86) regulations on the protection of laboratory animals. All efforts were made to minimize animal suffering and to reduce the number of animals used. Rat quadriceps were minced into small pieces using tweezers and scalpels in a Petri dish. For digestion and cell extraction, the tissue pieces were enzymatically digested using a solution of Collagenase type I (3 mg/mL) (Gibco, Invitrogen, Milano, Italy), and Dispase (4 mg/mL) (Gibco) prepared in phosphate-buffered saline (PBS, PH 7.2, Gibco), and incubated in a falcon tube overnight at 37˚C under stirring. To prevent bacterial contamination, gentamycin (0.2 mg/mL; Hospira, Milano, Italy) was added to the enzymatic solution. Isolated cells were separated from undigested pieces through a sterile filter (70 μm, Falcon), and the cellular suspension was centrifuged at 400 g for 7 min (Eppendorf Centrifuge). The pellet was washed with PBS, recentrifuged and re-suspended in Dulbecco's Modified Eagle Medium (DMEM) supplemented with Fetal Bovine Serum (FBS) (10% v/v) (Gibco, MA, Invitrogen), glutamine (1% v/v), penicillin-streptomycin (1% v/v), and Amphotericin B (1% v/v) (Lonza, Basel, Switzerland). The cells were seeded in a 35 mm tissue culture dish and maintained at 37˚C in a humidified atmosphere containing 5% v/v $CO_2$, and the medium was changed every 48 h. At confluence, the cells were harvested with trypsin/EDTA 0.2 mg/mL and re-seeded in appropriate tissue culture plates.

The phenotypic characterization of rat muscle-derived cells was performed based on the expression of myogenin as a muscle-specific biomarker using immunofluorescence.

The effect of hyaluronan was evaluated on rat muscle-derived cells by three different experimental set ups:

a. Physiological condition:
   Control samples (untreated cells) were compared to HHA, LHA, and HCC (treated cells) treatments used at a final concentration of 1.6 mg/mL w/w, on cell proliferation by Time-Lapse Video Microscopy (TLVM), and biomarkers expression by immunofluorescence and western blotting.

b. The oxidative stress *in vitro* model:
   We aimed at evaluating if the hyaluronan gels were able to:

1. counteract the detrimental effect of hydrogen peroxide stress- and in this case the cells were initially added with $H_2O_2$ (50 μM) for 30 min and successively treated with the HA gels (HHA, LHA, and HCC) at 1.6 mg/mL w/w.

2. prevent the damage of stressfull agent and in this case the cells were initially treated with the HA gels (HHA, LHA, and HCC) at 1.6 mg/mL w/w, and then exposed (challenged) with $H_2O_2$.

We called case 1) repair function; while case 2 is a prevention or protective function of the gels.

For 1 and 2 conditions, cell's metabolic activity was analyzed in time course until 72 h using the MTT assay.

Additionally, the antioxidant activity of the HAs was evaluated based on SOD-2 expression by performing western blotting after 24 h of treatment.

Finally, atrophy *in vitro* model was established to better evaluate the potentiality of hyaluronan gels to modulate cells response in this other damage model.

and c) Atrophy *in vitro* model

To establish an *in vitro* atrophy condition, rat muscle-derived cells were incubated with TNF-α (100 ng/mL) for 24 h and HHA, LHA, and HCC (1.6 mg/mL w/w). Specific atrophic transcription factors (FoxO3a and NF-kB), atrogenes (Atrogin and MURF-1-1), and desmin and myogenin, were assayed as specific muscle biomarkers, at the transcriptional and protein level using qRT-PCR and western blot, respectively.

**In vitro determination of cell morphology and proliferation using Time-Lapse Video Microscopy (TLVM).** The morphology and proliferation of rat muscle-derived cells were monitored and analyzed by time-lapse video microscopy station (TLVM) (Okolab, Naples, Italy). For these experiments, cells were seeded at a density of $2.5 \times 10^4$ in a 12-well plate, and treated with HHA, LHA, and HCC diluted in the medium at a final concentration of 1.6 mg/mL w/w. Quantitative data analysis was performed by calculating the cell number/cm$^2$ at different incubation times (0, 6, 12, 24, 48, 72, 96 h) in 4–5 fields of view for all samples tested.

**Immunofluorescence for the muscle-specific biomarker, myogenin.** In physiological conditions (experimental set up a), the basal expression of myogenin for characterization of muscle phenotype was analyzed using immunofluorescence. Specifically, $5.0 \times 10^3$ cells were grown on chamber slides (BD Falcon, Italy), and treated with HHA, LHA, and HCC. After 24 h, treated and the control cells were washed with PBS, fixed with paraformaldehyde 4% w/v, and permeabilized in 0.2% v/v Triton X-100 prepared in PBS. Immunofluorescence was performed using conditions reported previously [30]. Antibody against myogenin (diluted 1: 100; Abcam, Cambridge MA), was used as the primary antibody. FITC-conjugated goat anti-rabbit secondary antibody was used at a dilution of 1:1000 (Life Technologies, Milano, Italy). Nuclei were stained with 2'-(4-hydroxyphenyl)-5-(4-methyl-1-piperazinyl)-2,5'-bi-1H-benzimidazole trihydrochloride hydrate, bisBenzimide (Hoechst 0.5 mg/mL, Sigma-Aldrich, Milano, Italy). Images were captured using a fluorescence microscope Axiovert 200 (Zeiss) and analyzed using AxioVision 4.8.2.

**Atrogene and specific muscle protein evaluation by western blotting.** Western blots were performed as previously described [17–28]. Briefly, cells were lysed in RIPA buffer, and protein concentration was determined using the Bradford method. Twenty micrograms of intracellular proteins were loaded and resolved on a 10% SDS–PAGE gel. Antibodies against FoxO3a and NF-kB, atrogin, MuRF-1, myogenin, desmin, and SOD-2 (diluted 1:500 in T-TBS 0.1%, Santa Cruz Biotechnology, Dallas, TX) were used as primary antibodies. Horseradish peroxidase-conjugated donkey anti-mouse and goat anti-rabbit antibodies were used as secondary antibody (diluted 1:5000 in T-TBS 0.1; Santa Cruz Biotechnology, Dallas TX). The signal was detected using the ECL system (Chemicon-Millipore, Milano, Italy). Protein levels were normalized with respect to the expression of the housekeeping protein, actin (diluted 1:1000 in T-TBS 0.1%). The semi-quantitative analysis of protein levels was carried out by the Gel Doc 2000 UV System and the Gel Doc EZ Imager (Quantity one software, Bio-Rad Laboratories).

**MTT-test to evaluate cell viability in the presence of hyaluronan.** Cytotoxicity was assessed using $3.0 \times 10^4$ cells seeded in a standard 24-well culture plate, pre-treated with 50 μM

**Table 1. Primer sequences for real-time PCR.**

| Gene | GenBank accession no. | Primer sequence | Amplicon length (bp) |
|------|----------------------|-----------------|---------------------|
| GAPDH | NM_017008 | 5'-CATCCTGCACCACCAACTG-3' | 117 |
| | | 5'-CACAGTCTTCTGAGTGGCAG-3' | |
| FoxO3a | NM_001106395.1 | 5'-TCCCTGAAGGGAAGGAGC-3' | 105 |
| | | 5'-CTCGTCCAGGATGGCGTAG-3' | |
| Atrogin | NM_133521.1 _ | 5'-AAGCTTGTGCGATGTTACCC-3' | 110 |
| | | 5'-CCAGGAGAGAATGTGGCAGT-3' | |
| MuRF-1 | NM_080903 | 5'-CTCGCAGCTGGAGGACTCC-3' | 103 |
| | | 5'-CTCGTCCAGGATGGCGTAG-3' | |
| Myogenin | NM_017115.2 | 5'-CCTGCCCTGAGATGAGAGAG-3' | 106 |
| | | 5'- TGGAAGGTTCCCAATATCCA-3' | |
| Desmin | NM_022531.1 | 5'- AGC CTG GGT CAG AGA CAG AA-3' | 106 |
| | | 5'- TAT CTC CTG CTC CCA CAT CC-3' | |

$H_2O_2$ (30 min), and then treated with HHA, LHA, and HCC, respectively (1.6 mg/mL w/v) until 72 h. Analyses were performed after 24, 48, and 72 h post-treatment by measuring the reduction of the tetrazolium dye 3-(4,5-dimethylthiazol-2-yl)-2,5-diphenyltetrazolium bromide (MTT) [31]. The optical densities of the obtained solutions were measured at 570 nm using a Beckman DU 640 spectrometer (Beckman, Milano, Italy). The relative cell viability was calculated as a percentage of the maximal absorbance (vitality = $100 \times$ mean $OD_{treated\ cells}$/ mean $OD_{control}$).

**mRNA analyses using qRT-PCR.** For transcriptional analyses of atrogenes (FoxO3a, MuRF-1, and Atrogin), myogenin, and desmin, $5.0 \times 10^4$ cells in a 12 well were seeded. In order to perform qRT-PCR, total RNA was isolated following a previously described procedure [23]. The specific oligonucleotide sequences were reported in Table 1. The samples were analyzed in triplicate, and the mRNA expression of specific genes was normalized to the glyceraldehyde-3-phosphate dehydrogenase (GAPDH) housekeeping gene. The variations of gene expressions were evaluated using the quantification $2^{-\Delta\Delta Ct}$ method [32].

## Statistical analyses

The Student's t-test was used to determine statistically significant differences (p<0.01). The comparisons were accomplished with an average of three independent experiments, each in duplication to avoid possible variation of cell cultures. Also statistical significance for the data reported in Figs 3, 5A and 6A was determined using one-way ANOVA and Tukey post hoc test for comparing a family of five estimates by JASP soft-ware (https://jasp-stats.org) and reported in Supporting information.

## Results

### Hyaluronan effect on rat muscle-derived cells phenotype and cell growth

The phenotypic characterization of rat muscle-derived cells was performed by analyzing the expression myogenin as specific skeletal muscle marker through immunofluorescence staining (S1 Fig) revealed consistent phenotype of the cells like their source from skeletal muscle. The primary cells were cultivated and expanded to obtain the amount needed for the experiments, and in the second passage, time-lapse experiments were performed to evaluate the cell growth in the presence of HHA, LHA, and HCC. The cells were monitored for 96 h, and specific times of observation (0-6-12-24-48-72-96 h) were considered for data analyses. The growth curve is

shown in Fig 2A. The results demonstrated that HCC promoted cell proliferation faster than LHA and HHA after 24 h of treatment. In particular, HHA and LHA showed a slight increase in cell density (1.15 fold) as compared to the control (un-treated cells), while it was 2.3 fold in HCC. Subsequently, the phenotypic characterization of myogenin in the presence of hyaluronans revealed a higher expression of myogenin in the presence of HCC as compared to HHA and LHA treatments. However, all treatments were superior to the untreated control cells (Fig 2B). Densitometric analyses showed increased expression of both proteins myogenin and desmin in the HA treated samples (a 4fold increase was promoted by HCC treatment) in comparison with the untreated samples (Fig 2C).

## Effect of hyaluronan treatment on $H_2O_2$ treated skeletal muscle-derived cell: Cell's metabolic activity using colorimetric assay

The viability of cells stressed with $H_2O_2$ and successively added with HA gels (rescue/repair function) at three different times (24-48-72h) revealed a similar behavior of HHA and LHA treatment both increasing the cells' metabolic activity of approximately 55% respect to untreated cells and of 79% as compared to the $H_2O_2$ insulted cells at 72h. While HCC treated cells showed a cell's metabolic activity increase of approximately 83% respect to untreated cells and of 92% as compared to respective $H_2O_2$ insulted cells (p<0.001). HCC treatments also showed to prompt cell growth better then the HHA and LHA ones (Fig 3A). Alternatively, in samples pre-treated with HAs (protective function) and then stressed with $H_2O_2$ (Fig 3B), HCC preserved the viability by 2.5, and 1.5 fold higher than untreated cells and HHA treatment, respectively. HCC treatment also showed a significant increase in cell's metabolic activity of 57% and 85% at 48 and 72h as compared to $H_2O_2$ treatment (p<0.001). While HHA treatment increased the cells' metabolic activity approximately 48% and 37% and LHA treatment of 37% and 81% at 48 and 72 h, respectively, as compared to $H_2O_2$ stressed cells (p<0.05 and p<0.001). In addition, one-way anova test showed that there was a significance in HCC treatments respect all others for the different times and for both experimental conditions investigated. While HHA and HCC presented similar outcomes at 48h in the protective function.

At shorter incubation times, differences among the various treatments were less significant. These results proved that HA gels preserved cells from death counteracting the detrimental oxidative stress. Generally, treatments performed differently if they are compared in rescuing the cells from a stressful/damaged condition or as protective pre-treatments.

## SOD-2 expression

The analysis of the expression of SOD-2 protein through western blotting was done to test the efficacy of the HA in cell protection under oxidative stress. The densitometric analyses of the $H_2O_2$ pre-treated cells showed that all HA treatments reduced the SOD-2 expression (p<0.01). Both HHA and LHA showed a 1.5 fold reduction in comparison with $H_2O_2$ challenged cells (negative control), and HCC, with a 2 fold SOD-2 expression decrease, was identified to be most effective (Fig 4A). While in the experimental set-up with HA gels pre-treatment before the $H_2O_2$, the protective condition of the SOD-2 expression was also found to be reduced. Specifically, the protective effect of HHA was superior and significant compared to the other treatments (p<0.01), while HCC was significantly more efficient in rescuing the cells from stressful conditions. Overall, the effect of hyaluronan treatments was more marked when these were added after the oxidative stress (rescue condition).

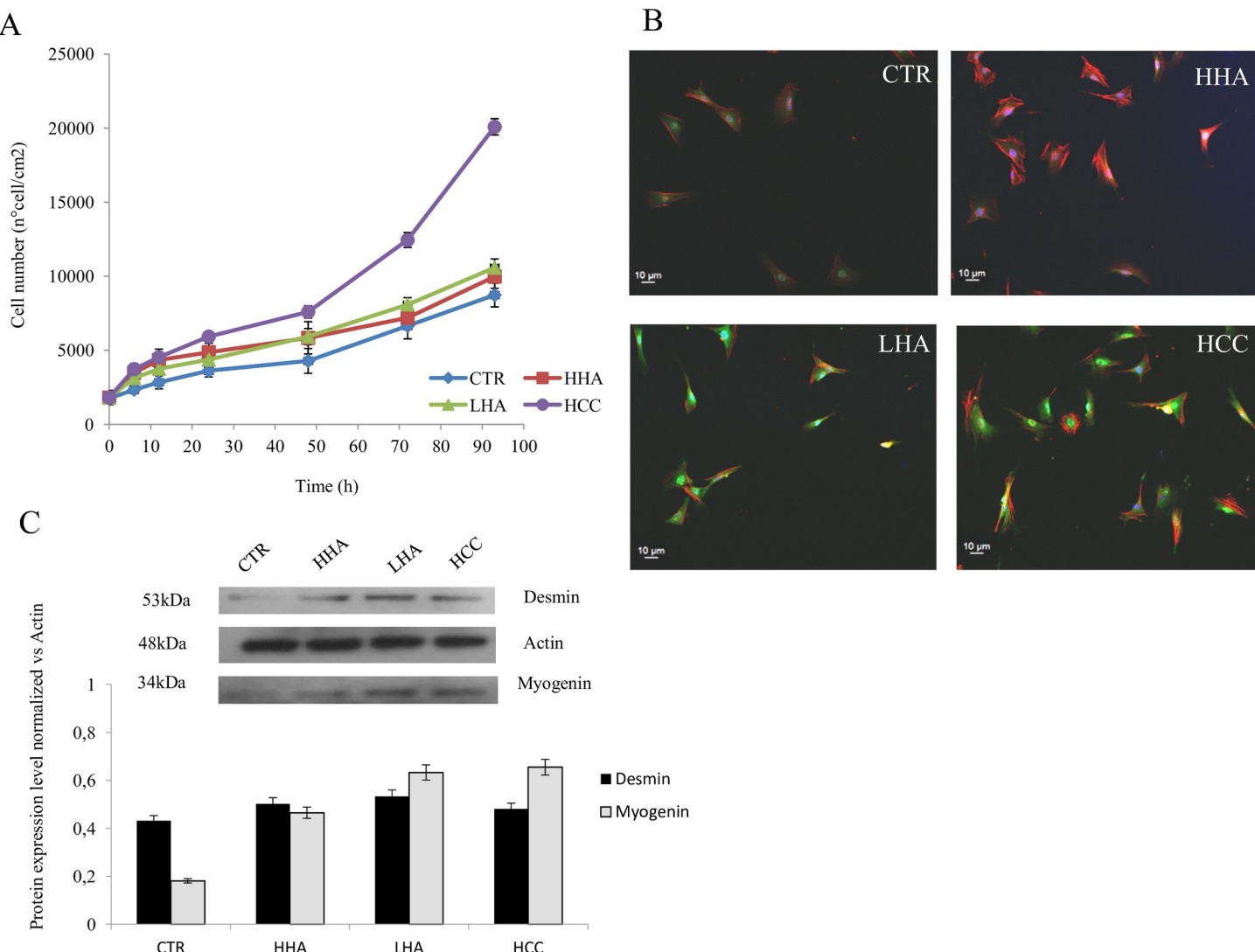

**Fig 2. Proliferation assay of muscle-derived cells performed by time-lapse experiments at 0, 6, 12, 24, 48,72, and 96 h of incubation with HHA, LHA, and HCC at 1.6 mg/mL w/w, compared to CTR.** (A) Data shown are means±SD of four fields of view of three different wells for each sample. Proliferation experiments were repeated three times. (B) Immunofluorescence staining of myogenin in the presence of HHA, LHA, and HCC. Blue nuclei, red actin cytoskeleton, green myogenin. (C) Western blotting analyses and densitometry. Desmin and myogenin protein levels in rat muscle-derived cells treated with HHA, LHA, and HCC for 24 h. Specific bands corresponding to the proteins of interest are measured using commercially available software (Image J software). Densitometric analyses of specific bands were obtained for both the proteins, normalized to actin expression. Data shown are average of duplicates and means ± SD. The statistical significance was analyzed through the students't-test (*p<0.01).

## Atrophy model specific atrophic biomarkers evaluated at transcriptional and protein level

In order to understand the specific molecular mechanisms as the basis of the cell damage, specific regulatory factors involved in atrophy signaling were examined, as reported in Fig 1. Specifically, the atrogenes (FoxO3a, Atrogin and MuRF-1) were analyzed at the transcriptional level using qRT-PCR (Fig 5A). All hyaluronan treatments showed a down-regulation of these biomarkers as compared to the positive control of atrophy (TNF-α) (p<0.001). However, FoxO3a was less modulated and slightly higher respect to TNF-α in the presence of HHA. In addition, HHA and LHA significantly reduced the expression levels of Atrogin and MuRF-1 in

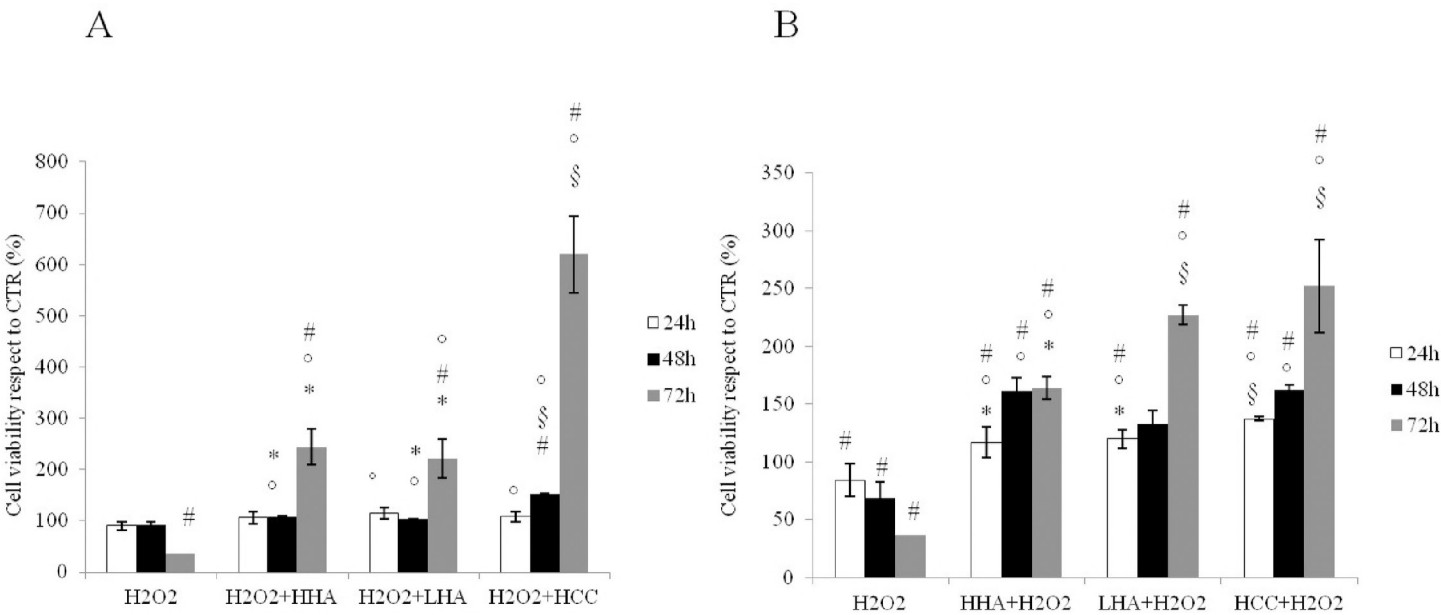

Statistical significance obtained by ANOVA test for comparing a family of 5 estimates : # p<0.05 or less vs untreated-cells (CTR); ° p< 0.05 or less vs $H_2O_2$, § p<0.05 or less vs HHA;* p< 0.05 or less vs HCC.

**Fig 3. MTT assay.** (A) Pre-treatment with $H_2O_2$ (50 μM) for 30 minutes and post -treatment with HHA, LHA and HCC (1.6 mg/mL w/w) for 24-48-72 hours (Repair function). (B) Pre-treatment with HHA, LHA and HCC (1.6 mg/mL w/w) for 24-48-72 hours and post-treatment with $H_2O_2$ (50 μM) for 30 minutes (Protective function). Data shown are average of triplicates and means ± SD. The statistical significance was analyzed through one-way ANOVA and Tukey post hoc test for comparing a family of 5 estimates: # p<0.05 or less vs untreated-cells (CTR); ° p< 0.05 or less vs $H_2O_2$, § p<0.05 or less vs HHA;* p< 0.05 or less vs HCC.

comparison with TNF-α. Moreover, the effect of HCC treatment was found to be the most efficient (p<0.001). It showed about 7 fold reduction in expression of FoxO3a while expression levels of Atrogin and MuRF-1 were reduced by 22, and 12.6 fold as compared with TNF-α, respectively. In addition, NF-κB and FoxO3a as up-stream activators of atrophy signaling were also evaluated (Fig 5B) through western blotting. A significant protein reduction was obtained when the cells were treated with HA. HHA reduced the expression levels of NF-κB, FoxO3a, Atrogin, and MuRF-1 (Fig 5B) by 1.2, 2.5, 5.5, and 6.3 fold, respectively, as compared to TNF-α (p<0.01). Whereas, the LHA treatment reduced the expression of NF-κB, and FoxO3a by 1.7 and 1.4 fold as compared to the negative control (un-treated cells), respectively, while that of Atrogin and MuRF-1were reduced by 1.4 fold, each as compared to TNF-α (Fig 5C) (p<0.01). The HCC treatments also reduced the expression levels of FoxO3a, NF-κB, Atrogin, and MuRF-1 by 1.4, 1.5, 1.9, and 6.5 fold compared to TNF-α.

## Analyses of myogenin and desmin at transcriptional and protein level

The gene expression analysis of myogenin and desmin showed increased expression levels of both the proteins for all HA treatments as compared to TNF-α atrophic cells. In particular, the expression level of desmin was significantly increased by 7 fold and myogenin by 4 fold in HCC treated cells (Fig 6A) (p<0.001). Moreover, western blotting analyses corroborated the mRNA results, where the desmin and myogenin genes were more expressed in HA treated cells as compared to TNF-α. Specifically, LHA and HCC significantly upregulated the expression of desmin by 2.2 and 2.3 fold, respectively, as compared to TNF-α, while for myogenin expression, both LHA and HCC increased its level by 1.6 and 1.75 fold, respectively (Fig 6B) (p<0.01).

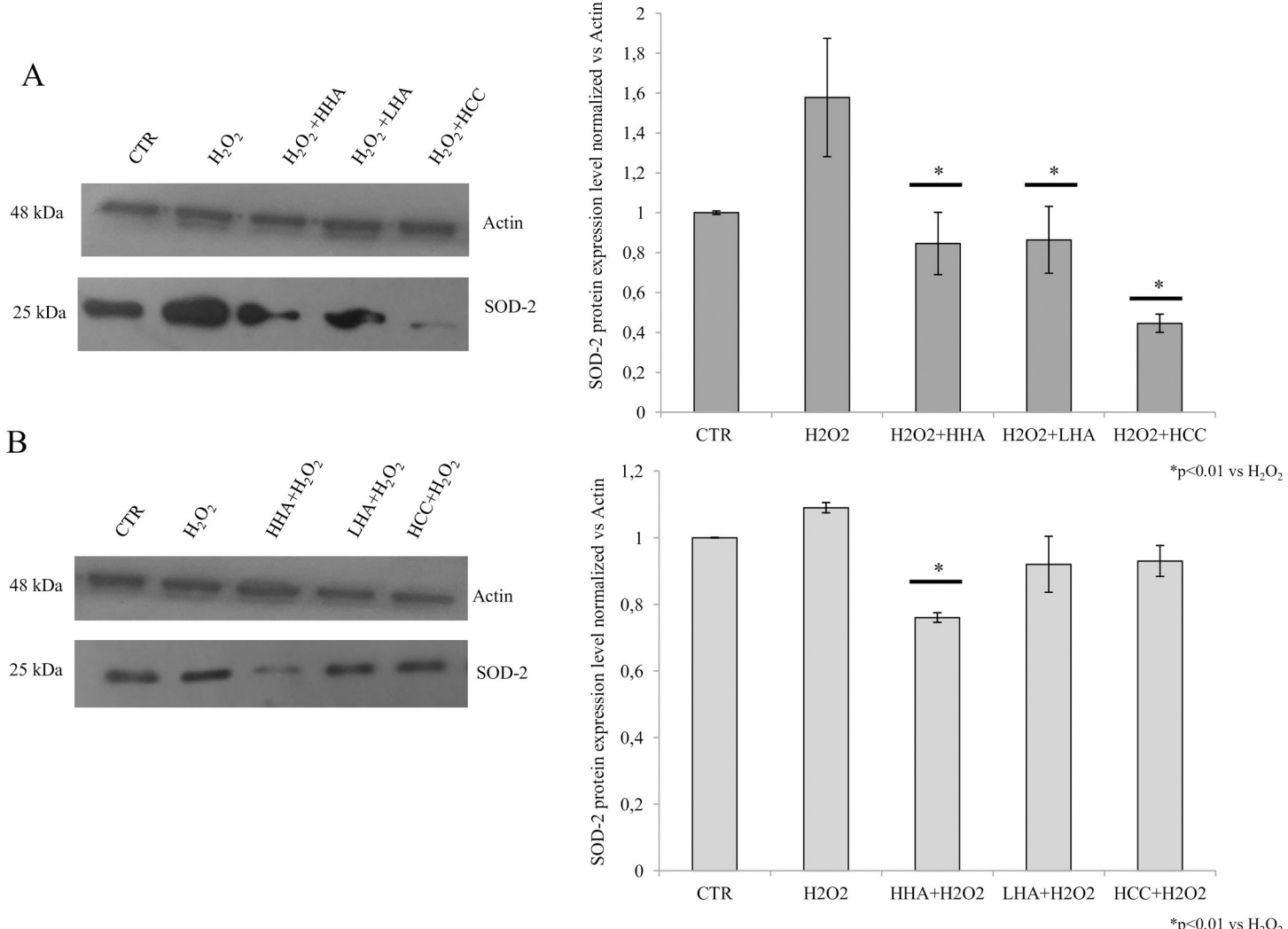

**Fig 4. Western blotting analyses and densitometry of SOD-2.** The upper bands show the expression of the actin housekeeping protein, while the lower bands indicate the SOD-2 expression. The histograms indicate the densitometric normalized analysis on actin band for each of the cell extracts (control and treatments). (A) Pre-treatment with $H_2O_2$ (50 μM) for 30 minutes and post -treatment with HHA, LHA and HCC (1.6 mg/mL w/w) for 24-48-72 hours (Repair function). (B) Pre-treatment with HHA, LHA and HCC (1.6 mg/mL w/w) for 24-48-72 hours and post-treatment with $H_2O_2$ (50 μM) for 30 minutes (Protective function). Data shown are average of duplicates and means ± SD. The statistical significance was analyzed through the students't-test (*p<0.01) respect to $H_2O_2$ treated cells.

## Discussion

In the present study, we demonstrated that TNF-α induced *in vitro* atrophy in rat muscle-derived cells by activating atrogenes (e.g., Atrogin and MuRF-1) expression through FoxO3a and NF-kB transcription factors pathway and negatively modulating the muscle phenotype-specific proteins, such as myogenin and desmin. Furthermore, hyaluronan gels, showed a higher potential to rescue $H_2O_2$-challenged cells, as previously verified in other *in vitro* models [28, 33]. Our results clearly showed that hyaluronan gel-based treatments supported rat skeletal muscle primary cell growth, preserved the specific phenotype counteracting the correlated cell degeneration process. Specifically, of the different hyaluronans tested, the effects of HCC were significantly higher in comparison to the linear HA of high and low molecular weight.

The quantitative analyses performed by TLVM proved that HCC promoted muscle cells proliferation, doubling the density compared to both the linear HAs and the control. Earlier

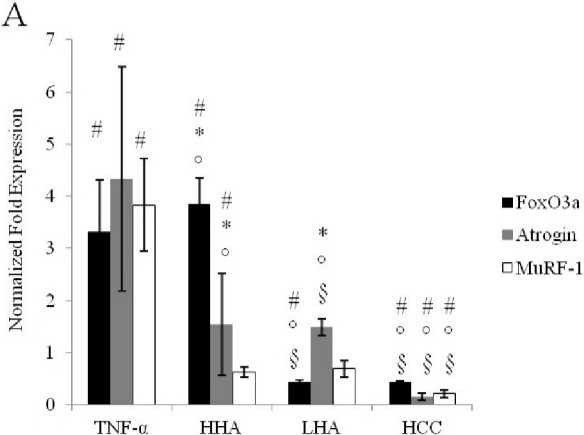

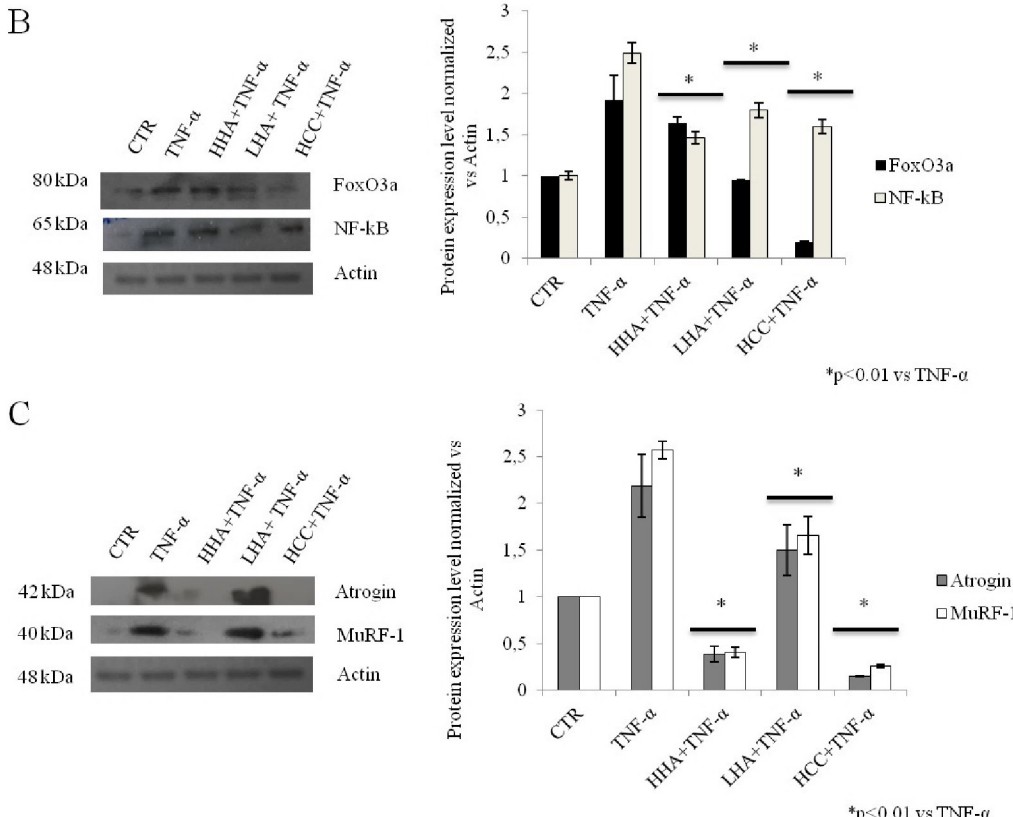

**Fig 5. Atrophy *in vitro* model.** (A) Quantitative Real-Time PCR of atrogenes. RNA expression was determined by quantitative real-time PCR on rat muscle-derived cells treated with TNF-α along HHA, LHA, HCC at 1.6 mg/mL w/w for 24 h. Data shown as average of triplicates and each column represents the mean and error bars indicating the standard deviation. The statistical significance was analyzed through one-way ANOVA and Tukey post hoc test for comparing a family of 5 estimates: # p<0.05 or less vs untreated-cells (CTR); ° p< 0.05 or less vs TNF-α, § p<0.05 or less vs HHA;* p< 0.05 or less vs HCC. Western blotting analyses of (B) NF-kB and FoxO3a transcription factors and (C) atrogenes were performed and normalized to Actin housekeeping protein. Data shown are average of duplicates and means ± SD. The statistical significance was analyzed through the students't-test (*p<0.01) respect to TNF-α treated cells.

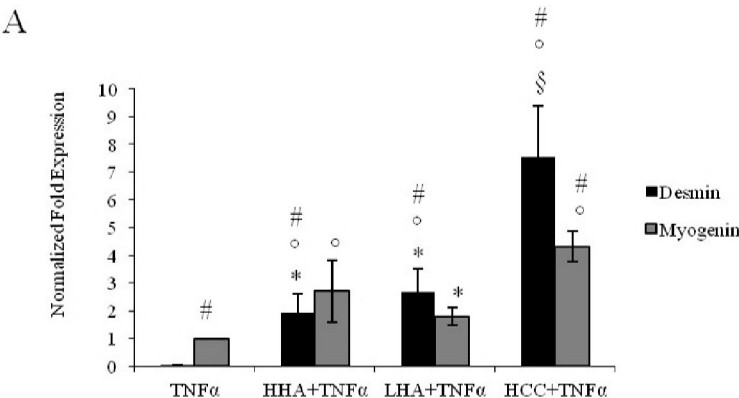

Statistical significance obtained by ANOVA test for comparing a family of 5 estimates: # p<0.05 or less vs untreated-cells (CTR); ° p< 0.05 or less vs TNF-α , § p<0.05 or less vs HHA;* p< 0.05 or less vs HCC.

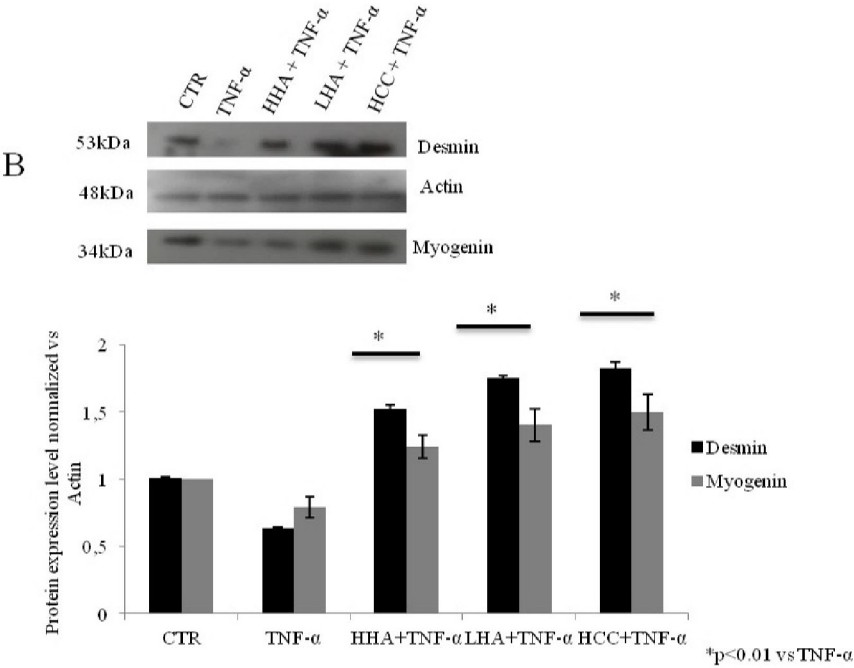

**Fig 6. Desmin and myogenin were analyzed at transcriptional and protein levels in rat muscle-derived cells treated for 24 h with TNF-α and HHA, LHA, HCC at 1.6 mg/mL w/w, each.** (A) qRT-PCR was conducted to analyze the gene expression levels. The statistical significance was analyzed through one-way ANOVA and Tukey post hoc test for comparing a family of 5 estimates: # p<0.05 or less vs untreated-cells (CTR); ° p< 0.05 or less vs TNF-α, § p<0.05 or less vs HHA;* p< 0.05 or less vs HCC. (B) Western blotting analyses of specifics bands obtained for both protein expressions are normalized to actin expression. Data shown are average of duplicates and means ± SD. The statistical significance was analyzed through the students't-test (*p<0.01) respect to TNF-α treated cells.

studies have reported that exogenous hyaluronan is able to modulate muscle precursor cell proliferation, migration, and differentiation [33]. It has also been reported that HA may play a role in binding collagen fibers, thus affecting muscle elasticity. This mechanism could be related to the concentration of HA in the fasciae [34]. Specific proteins, such as desmin and myogenin, were also implicated in tissue repair and regeneration [35,36]. Our data demonstrated that these two proteins are better modulated in the cells treated with HCC both in physiological and atrophic conditions. Recent studies reported the need to find targeted treatments

to promote skeletal muscle tissue growth and functionality by increasing protein synthesis (or decreasing protein degradation) defining specific regulation pathways as well as the myogenesis signaling [37,38]. So, in the present study, we have investigated the key transcription factors of muscle atrophy in order to elucidate its underlying mechanism. It has also been reported that the activation of FoxO3a and NF-kB, promotes the expression of Atrogin and MuRF-1, which is directly correlated to the loss of muscle mass and function [39,6]. As expected, the expression levels of atrophic proteins were increased in the presence of TNF-α while significantly decreased when the cells were treated with hyaluronans. In particular, the HCC was most efficient at improving muscle cell functionality. Recently Miky et al. (2019) [7] have shown that the pro-inflammatory signal related to the atrophy process impaired the cell function, increasing the atrogenes factors expression, thus, inhibiting the atrophic pathway and restoring the myogenic function [40]. Moreover, the role of oxidative stress in atrophic muscle due to an imbalance between the cellular antioxidant systems and reactive oxygen species (ROS) production has been reported [34,11]. It has also been shown that HA can protect the cells against damage either by exerting antioxidant activity to scavenge ˙OH, peroxynitrite (ONOO–), $O_2^-$ and peroxyl radicals in a dose-dependent manner, either through the chelation or elimination of $Fe^{2+}$ or by deletion of the proteins bound to $Fe^{2+}$ [26, 34]. In this respect, the $H_2O_2$ challenge was used in the present study, HCC, beyond the beneficial effect of linear HA increased cell viability and proliferation in both oxidative stress conditions. The activity of antioxidant enzymes, such as SOD-2, has also been studied in myocytes [9–12]. In our experiments, HCC also showed an extensive reduction of SOD-2 protein, proving that it is more efficient in promoting the rescue effects. Considering the recent literature, HA function in muscle cells may possibly involve the Tumor necrosis factor (TNF)-stimulated gene 6 (TSG-6). In fact, TSG6 is able to interact with HA [41–43], activating in cell response a variety of growth factors and pro-inflammatory mediators. Milner and collaborators (2003) [41], proposed a high TSG6 expression level in skeletal muscle. For this reason we may suppose that beside the CD44 mediated signaling also TSG6 may be involved in the molecular interaction mechanism responsible for the biological effect. In this respect it will be of great interest a specific assessment of the HCC interaction in comparison to the sole high and low molecular weight fraction, not only in the muscle cell line, but also in other *in vitro* model (e.g. OA).

Overall, this study demonstrated that hyaluronan and, in particular, hybrid cooperative complexes made by high and low molecular weight chains of HA inhibited or counteracted the biochemical pathways involved in cell stress and damage preserving viability, phenotype, and overall functionality. Considering all the results together, we can suggest that hyaluronan based treatments provided an *in vitro* evidence of their potentiality in the therapeutic approach to treat muscle atrophy.

## Conclusions

In this study, the preserving, beneficial, and protective effect of different hyaluronan-based gels on the recovery of stressed cells and muscle atrophy was evaluated and validated in an *in vitro* model based on rat muscle-derived cells. The two linear hyaluronans at high and low molecular weight proved their positive effect in modulating stress conditions. In addition, our results showed that HCC has a greater potential in preserving the muscle phenotype, promoting cell proliferation, and reducing the reactive oxygen species damage and atrophic biomarkers expression. However, further *in vivo* studies are necessary to confirm the effectiveness of HCC as potential new ingredients/active principles in the treatment of skeletal muscle disorders.

## Supporting information

**S1 Fig.**
(PDF)

**S2 Fig.**
(DOCX)

**S3 Fig.**
(DOCX)

**S4 Fig.**
(DOCX)

**S1 Raw images.**
(PDF)

## Acknowledgments

The authors would like to thank Editage for English language editing.

## Author Contributions

**Conceptualization:** Antonietta Stellavato, Barbara Rinaldi, Chiara Schiraldi.

**Data curation:** Antonietta Stellavato, Lucrezia Abate, Valentina Vassallo, Maria Donniacuo, Barbara Rinaldi, Chiara Schiraldi.

**Funding acquisition:** Chiara Schiraldi.

**Investigation:** Antonietta Stellavato, Lucrezia Abate, Valentina Vassallo, Maria Donniacuo.

**Methodology:** Antonietta Stellavato, Lucrezia Abate, Valentina Vassallo, Chiara Schiraldi.

**Supervision:** Chiara Schiraldi.

**Validation:** Antonietta Stellavato, Chiara Schiraldi.

**Writing – original draft:** Antonietta Stellavato, Barbara Rinaldi.

**Writing – review & editing:** Chiara Schiraldi.

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
