## [Decision Letter · Decision Letter 0]

9 Apr 2020

PONE-D-20-03354

An in vitro study to assess the effect of hyaluronan-based gels on muscle-derived cells: Highlighting a new perspective in regenerative medicine

PLOS ONE

Dear Prof Schiraldi,

Thank you for submitting your manuscript to PLOS ONE. After careful consideration, we feel that it has merit but does not fully meet PLOS ONE’s publication criteria as it currently stands. Therefore, we invite you to submit a revised version of the manuscript that addresses the points raised during the review process.

The study is well written and interesting, some aspect should be added, in particular some comments on the machinery which could be involved as well some details should be more explained in the text.

We would appreciate receiving your revised manuscript by May 24 2020 11:59PM. To enhance the reproducibility of your results, we recommend that if applicable you deposit your laboratory protocols in protocols.io, where a protocol can be assigned its own identifier (DOI) such that it can be cited independently in the future. For instructions see: http://journals.plos.org/plosone/s/submission-guidelines#loc-laboratory-protocols

We look forward to receiving your revised manuscript.

Kind regards,

Alberto G Passi, MD PhD

Academic Editor

PLOS ONE

Journal Requirements:

Additional Editor Comments (if provided):

The Authors presented a very interesting study on the role of different hyaluronan on the rescue of primary cells under stress and in recover of muscle atrophy. The study is based on the comparison of different hyaluronan polymers with pharmaceutical grade including high (HHA), low molecular weight (LHA) hyaluronan polymer, including a hybrid cooperative complexes (HCC) of HA. The HCC is demonstrated to play an intriguing role, in fact the data indicate that cells increased specific markers and growth. Control experiments support the protective role of HCC compared to other supports.

The study is well written and interesting, nevertheless reviewers raised some points that should be addressed as some information about the mechanistic aspects of the difference between HCC and other HAs, some infoprmation on the receptors or TASG6 involvement and more details in the methods in order to clarify the aspects discussed in the text.

Reviewers' comments:

Reviewer's Responses to Questions

**Comments to the Author**

1. Is the manuscript technically sound, and do the data support the conclusions?

Reviewer #1: Yes

Reviewer #2: Partly

2. Has the statistical analysis been performed appropriately and rigorously? 

Reviewer #1: Yes

Reviewer #2: No

3. Have the authors made all data underlying the findings in their manuscript fully available?

Reviewer #1: Yes

Reviewer #2: Yes

4. Is the manuscript presented in an intelligible fashion and written in standard English?

Reviewer #1: Yes

Reviewer #2: Yes

5. Review Comments to the Author

Reviewer #1: The authors describe the effects of HA of high and low mol. mass and conjugated HA on rat muscle derived cells. Authors studied cell proliferation, viability and in vitro atrophy finding that conjugated HA had the better effects.

The manuscript is clearly written and could be of interest in several clinical applications.

Several points, however, need to be addressed by the authors to increase discuss and to hypothesize the molecular mechanisms that trigger such HA treatments.

Have the authors an hypothesis of the mechanism of HCC? could HCC uses similar receptors of LHA and HHA?

Are the different HA added stable in the cell medium or such HA are degraded (a measurement of HYALs could be usefull to discuss this point)?

Many function of HA are know to be mediated by HC-HA complexes formed by TSG6. Have the authors investigated whether or not TSG6 could be involved?

As HCC is patented by one of the authors, is there any conflict of interest?

Reviewer #2: The manuscript demonstrates that a composite of high and low molecular weight hyaluronan (HCC) better preserves muscle cell phenotype than high molecular weight hyaluronan (HHA) or low molecular weight hyaluronan (LHA) in preserving phenotype of primary muscle cells and protect them from atrophy. While this finding is interesting, there are notable details lacking from the methods (e.g., description of controls) and some questions about the data that make it difficult to interpret. Details are provided in the comments below.

1. Please add an “and” before “c)” in the 3rd sentence to complete this list.

2. The abstract states: “The results showed that HCC and HHA increased cell proliferation by 1.15 and 2.3 folds in comparison to control, respectively.” Please clarify what the control is here in the text. It is also not clear from the Methods or Results text what the control conditions are.

3. The abstract states: “In this model, HCC revealed a noteworthy beneficial effect on the myogenic biomarkers indicating that it could be used as a promising platform for tissue regeneration with specific attention to muscle cell protection against stressful agents.” However, text up to this point indicates that HHA worked at least as well, if not better, than HCC.

4. Introduction, pg. 4: “HA is a hygroscopic molecule that is able to synthesize the ECM…” The use of “synthesize” is not appropriate here. The HA does not produce the ECM, although it can act to organize it structurally by complexing with other ECM macromolecules.

5. Method, Pg. 8, paragraph 3- Please provide more information on the fluorescence microscope such as what brand and model was used.

6. Method, pg.9, paragraph 2- Authors state that “Cytotoxicity was assessed using 3.0 × 104 cells seeded in a standard 24-well culture plate, pre- treated with 50 μM H2O2 (30 min), and then treated with HHA, LHA, and HCC, respectively (0.16% w/v) for 24 h. Analyses were performed after 24, 48, and 96 h post-treatment by measuring the reduction of the tetrazolium dye 3-(4,5-dimethylthiazol-2-yl)-2,5-diphenyltetrazolium bromide (MTT) “. Does this mean that HA was kept in the medium for only the first 24 hrs? Later, in the Results on pg. 12, the text implies that instead the times of HA treatments were different. Please clarify this issue.

7. How was the concentration of HA controlled for because HHA, LHA and HCC experimental conditions?

8. Were different concentrations of HA in each solution evaluated? It will be important to know if these effects are concentration dependent.

9. Please add indications onto graphs in Figures to show where data are significantly different from each other. This has only been done in Figures 3 and 6. It would also help to state when findings are statistically significant when discussing them in the Results text.

10. Figure 1 is never referred to in the text.

11. Table 1, pg.10.- Table does not look very polished: e.g. some Genbank accession no are underlined while others are not, some rows are wider than others, etc.

12. Results: The subtitles like “(experimental Set-up a)” are not really necessary.

13. This may have happened during pdf conversion, but all of the figures have blurry lines and text.

14. In Figure 2B, the images require scale bars.

15. The procedures for hydrogen peroxide “pre-challenge” and “post-challenge” are not clear from the methods or figure 4 caption.

16. As the MTT assay does not explicitly measure numbers of viable cells, it is more appropriate to refer to MTT results as reflective of the cells’ metabolic activity.

17. Statistical analyses: Students t-tests are not appropriate for these data sets where multiple comparisons are being made or where multiple independent variables are evaluated (e.g., Fig. 3, where time and treatment are variables). Please also include the software used for statistical analysis in the Methods.

18. Results: For all graphs, it is not clear what the asterisks indicating significance are comparing to here. For example, in Fig. 5A, is HHA MuRF-1 different than HHA FoxO3a? For LHA and HCC, do the different genes have different expression than the other genes or are all genes different than TNFalpha?

19. Results: For all figures, please indicate what the error bars represent (e.g., Standard deviation/ SEM?) in the captions.

20. Results: For all figures, please indicate what the number of replicates used for data analysis in the captions.

21. Results, Fig 3: Was there a positive control for this experiment?

22. Result, Fig 4: What are the negative and positive controls for this experiment?

23. Results, Fig 5. (B, C): The actin control rows appear to be identical in both western blot figures. Were these all run on the same blot? Also, the HCC treatment condition appears to have more actin. Is there a reason for this?

24. Results, Fig 6(B). For western blotting sample, random hyphens were added to the LHA and HHA the abbreviations.

25. Discussion, pg. 17: Please be consistent in italic style of “in vitro”.

26. Supporting information, S1B: Background in the red channel seems pretty high.

6. PLOS authors have the option to publish the peer review history of their article (what does this mean?). If published, this will include your full peer review and any attached files.

Reviewer #1: No

Reviewer #2: Yes: Stephanie Seidlits

---

## [Author Response · Author response to Decision Letter 0]

23 May 2020

Dear Editor,

Thank you for your consideration for our manuscript.

Hereby we submit the revised version of the manuscript number PONE-D-20-03354

“An in vitro study to assess the effect of hyaluronan-based gels on muscle-derived cells: Highlighting a new perspective in regenerative medicine” PLOS ONE

We considered the reviewer suggestions and the manuscript was modified and corrected accordingly. 

All the minor points were addressed and modified in the text, the manuscript with track changes revisions (revised manuscript with track changes) is provided beside the “clean”version (“manuscript” without tracked changes) according to the PlosOne author guidelines.

The original uncropped and unadjusted images underlying all blot results are reported in Supporting information files.

Please find following the response to reviewers:

Response to Reviewer #1: 

5. Review Comments to the Author

Reviewer #1: The authors describe the effects of HA of high and low mol. mass and conjugated HA on rat muscle derived cells. Authors studied cell proliferation, viability and in vitro atrophy finding that conjugated HA had the better effects.

The manuscript is clearly written and could be of interest in several clinical applications.

Several points, however, need to be addressed by the authors to increase discuss and to hypothesize the

molecular mechanisms that trigger such HA treatments.

Have the authors an hypothesis of the mechanism of HCC? could HCC uses similar receptors of LHA and HHA?

HCC is based on the cooperativity of hydrogen bonding among che high molecular weight Hyaluronan chains and the low molecular weight ones. These are thermally stabilized. However, when diluted in the cell media, or in physiological environment the competition of solvent is becoming more and more effective in time releasing both the low and high molecular weight chains. For this reason, it is possibly the direct interaction of the slowly released single chains to release the effect. A study focus on the direct interaction of the complex on the cell wall specific receptors was not carried out yet. However, we think that the suggestion of the editor/referee is very interesting starting point for in depth analyses of the unraveled features of these HCC.

Are the different HA added stable in the cell medium or such HA are degraded (a measurement of HYALs could be usefull to discuss this point)?

HCC are more stable than HHA to hyaluronidases, specifically 80% of the initial molecular population with MW higher than 1 MDa was digested in 24h in presence of hyaluronidase (BTH 0.5 U/mL at 37°C) while 5% only was degraded of the same fraction in HCC (D’Agostino et al. 2015). Very few studies on the hyaluronidases synthesis in vitro are presented in literature, however the most recent is addressing this point regarding mast cells (Calve et al. J Biol Chem. 2019 Jul 26;294(30):11458-11472). In previous research paper we evaluated in a similar time scale the effect of HCC compared to HHA and LHA and generally the medium (containing the gel preparation) was exchanged every 48h. 

Many function of HA are know to be mediated by HC-HA complexes formed by TSG6. Have the authors

investigated whether or not TSG6 could be involved?

TSG6 has been demonstrated to interact with HA (Spinelli et al., Milner et al., and Baranova et al.). 

Caroline and collaborators propose a high expression level in skeletal muscle for this reason we may suppose that among the molecular mechanism (some of which mainly related to CD44) responsible for the biological effect also TSG6 may be involved. In this respect it will be of great interest a specific assessment of the HCC interaction in comparison to the sole high and low molecular weight fraction, not only in the muscle cell line, but also in other in vitro model (e.g. OA). The authors have been discussed potential function of TSG-6 referring our study in the discussion section (page 17 line 607-615).

As HCC is patented by one of the authors, is there any conflict of interest?

Prof Schiraldi is among the inventors, but the assignee is a company, since the research work was developed within a public-private joint project partially financed by Campania regional government (measure for innovation). We may add a specification in the section conflict of interest to specify this.

Response to Reviewer #2: 

Reviewer #2: The manuscript demonstrates that a composite of high and low molecular weight hyaluronan (HCC) better preserves muscle cell phenotype than high molecular weight hyaluronan (HHA) or low molecular weight hyaluronan (LHA) in preserving phenotype of primary muscle cells and protect them from atrophy. While this finding is interesting, there are notable details lacking from the methods (e.g., description of controls) and some questions about the data that make it difficult to interpret. Details are provided in the comments below.

1. Please add an “and” before “c)” in the 3rd sentence to complete this list.

 We added “and” before “c)” in the 3rd sentence to complete the list (page 8 line 163).

2. The abstract states: “The results showed that HCC and HHA increased cell proliferation by 1.15 and 2.3 folds in comparison to control, respectively.” Please clarify what the control is here in the text. It is also not clear from the Methods or Results text what the control conditions are.

In this sentence (page 2 line 38) we replaced “control” with “un-treated cells”. Also, we clarified the description of all the control conditions used for the three experimental set-up investigated from the Methods and Results text.

3. The abstract states: “In this model, HCC revealed a noteworthy beneficial effect on the myogenic biomarkers indicating that it could be used as a promising platform for tissue regeneration with specific attention to muscle cell protection against stressful agents.” However, text up to this point indicates that HHA worked at least as well, if not better, than HCC.

Yes, this is the meaning we wanted to give (page 2 line 40).

However, in figure 5 A, B and C a significant better effect from HCC was found in reducing atrogens expression. Results relative to oxidative stress (Fig 3A) and atrophy (fig 5C) HCC and HHA presented similar behavior counteracting stressful conditions. We underline that HHA resulted slightly more effective than HCC in reducing SOD-2 expression during protective effect (fig4B).

4. Introduction, pg. 4: “HA is a hygroscopic molecule that is able to synthesize the ECM…” The use of “synthesize” is not appropriate here. The HA does not produce the ECM, although it can act to organize it structurally by complexing with other ECM macromolecules.

We apologize, the sentence was not correct, we modified that inserting:

“HA is a hygroscopic molecule that is able to structurally organize the ECM by complexing with other ECM macromolecules”, page 4 line 78-79 in the introduction section.

5. Method, Pg. 8, paragraph 3- Please provide more information on the fluorescence microscope such as what brand and model was used.

Immufluorescence analyses was accomplished using an Axiovert 200 (Zeiss) microscope, detail have been added to the materials and methods section page 9 line 189-190.

6. Method, pg.9, paragraph 2- Authors state that “Cytotoxicity was assessed using 3.0 × 104 cells seeded in a standard 24-well culture plate, pre- treated with 50 μM H2O2 (30 min), and then treated with HHA, LHA, and HCC, respectively (0.16% w/v) for 24 h. Analyses were performed after 24, 48, and 96 h post-treatment by measuring the reduction of the tetrazolium dye 3-(4,5-dimethylthiazol-2-yl)-2,5-diphenyltetrazolium bromide (MTT) “. Does this mean that HA was kept in the medium for only the first 24 hrs? Later, in the Results on pg. 12, the text implies that instead the times of HA treatments were different. Please clarify this issue.

Thanks for the comment, we have clarified the description of the procedure performed in the methods section page 9 line 207 and modified the outcomes considering the metabolic activity in the results section line starting from page 13 line 283-298 (fig 3A and B).

7. How was the concentration of HA controlled for because HHA, LHA and HCC experimental conditions?

Respect to the initial gels, that were prepared from pharma grade powder at 16 g/L and thermally treated for 12 min at 120°C in the autoclave, we diluted them into the cell medium 1:10 to achieve a final concentration of 1.6mg/mL. For HCC we used a sterile syringes from IBSA Italia (IBSA Farmaceutici Italia Srl, Italy) diluted 1:20 into the cell medium, as reported in materials and methods section page 5 line 115.

8. Were different concentrations of HA in each solution evaluated? It will be important to know if these effects are concentration dependent.

No, we have used only one concentration (1.6 mg/mL) for all hyaluronans tested.

9. Please add indications onto graphs in Figures to show where data are significantly different from each other. This has only been done in Figures 3 and 6. It would also help to state when findings are statistically significant when discussing them in the Results text.

As requested by the referee we have performed t- student test analyses on all figures as suggested, and modified accordingly the figures and texts.

10. Figure 1 is never referred to in the text.

We have mentioned the figure 1 referring to atrophy signaling at page 3 line 59 of the introduction section.

11. Table 1, pg.10.- Table does not look very polished: e.g. some Genbank accession no are underlined while others are not, some rows are wider than others, etc.

According to the suggestion the table 1 was formatted according to plos one guidelines and inserted in the manuscript.

12. Results: The subtitles like “(experimental Set-up a)” are not really necessary.

We deleted subtitles as suggested by the referee. 

13. This may have happened during pdf conversion, but all of the figures have blurry lines and text.

We have upload figures as a 300 dpi following the journal guidelines however, we will up-load those again hoping that their quality will be improved. 

14. In Figure 2B, the images require scale bars.

We apologize, on the original figure scale bars were present and visible. The issue may be due to the pdf conversion. We will try to re-upload the image to make it better visible.

15. The procedures for hydrogen peroxide “pre-challenge” and “post-challenge” are not clear from the methods or figure 4 caption.

Ok, we have modified the description for oxidative stress in the materials and methods page 7 from line 150-158.

We aimed at evaluating if the hyaluronan gel were able to:

1) counteract the detrimental effect of Hydrogen peroxide treatment- and in this case the cells were initially challenged with H2O2 and successively treated with the gels.

2) To prevent the damage of stressfull agent- and in this case the cells were initially treated with the gels and then exposed (challenged) with H2O2.

We called case A) Rescue (repair function); while case B is a prevention/alert or protective function of the gels. 

16. As the MTT assay does not explicitly measure numbers of viable cells, it is more appropriate to refer to MTT results as reflective of the cells’ metabolic activity.

We agree with the referee that the figure axis may be misleading to the readers for this reason we modified the text indicating MTT results as cells’ metabolic activity page 13 line 283-298.

17. Statistical analyses: Students t-tests are not appropriate for these data sets where multiple comparisons are being made or where multiple independent variables are evaluated (e.g., Fig. 3, where time and treatment are variables). Please also include the software used for statistical analysis in the Methods.

We have used Microsoft excel in order to perform t-student test analyses. In particular, we compared the efficacy of different HAs treatments in counteracting H2O2 effect. For all time investigated (24, 48 and 72h) all HAs tested were effective respect to H2O2 treatment. However, the effect of HCC is more noticeable at 72h in the rescue/repair function (fig3A). Also, the HAs treatments resulted significant (p<0.01) with respect to H2O2 treatment at 48 and 72h in the protection function condition (Fig.3B). 

18. Results: For all graphs, it is not clear what the asterisks indicating significance are comparing to here. For example, in Fig. 5A, is HHA MuRF-1 different than HHA FoxO3a? For LHA and HCC, do the different genes have different expression than the other genes or are all genes different than TNF-α?

For figure 5A statistical analyses (t-test *p<0.01) were performed for each gene of different HAs treatment respect to TNF-α insult. Our purpose was to demonstrate that TNF-α insult induces atrogenes up-regulation and HAs treatment was effective in reducing it. HAs counteract the detrimental effect of TNF-α that is recognized as a treatment resembling atrophic conditions (Cho et al. 2018 ref 28 in the present manuscript).

19. Results: For all figures, please indicate what the error bars represent (e.g., Standard deviation/ SEM?) in the captions.

As requested, we indicated the error bars as standard deviation in the figure captions.

20. Results: For all figures, please indicate what the number of replicates used for data analysis in the captions.

We indicated the number of replicates used for our data analyses in the figure captions.

21. Results, Fig 3: Was there a positive control for this experiment?

The positive control (of stress) was the H2O2 treated cells while the negative control was the untreated-cells.

Untreated cells are the control

Stressed cells are the detrimental conditions.

On the already stressed cells were run to evaluate the biological effect of HAs.

22. Result, Fig 4: What are the negative and positive controls for this experiment?

The positive control (of stress) was the H2O2 treated cells while the negative control was the untreated-cells.

Untreated cells are the control

Stressed cells are the detrimental conditions.

On the already stressed cells were run to evaluate the biological effect of HAs.

23. Results, Fig 5. (B, C): The actin control rows appear to be identical in both western blot figures. Were these all run on the same blot? Also, the HCC treatment condition appears to have more actin. Is there a reason for this?

It is in fact a single western blot where we have separated the genes according to their function. For all experiment we have loaded the same protein amount (10�g tot) for all biomarkers investigated and also for actin. In this respect, actin signal is used as housekeeping protein as internal control for normalization.

24. Results, Fig 6(B). For western blotting sample, random hyphens were added to the LHA and HHA the

abbreviations.

Thank you for the remark, a mistake was made. We changed the name of samples (in the results section fig 6.B).

25. Discussion, pg. 17: Please be consistent in italic style of “in vitro”.

We controlled the italics for “in vitro” throughout the document.

26. Supporting information, S1B: Background in the red channel seems pretty high.

Ok, we have re-acquired the image by lowering the phalloidin red channel, for figure S1B.

We thank the reviewers for their suggestions that greatly improved our manuscript.

Kind regards

Chiara Schiraldi

Antonietta Stellavato

---

## [Decision Letter · Decision Letter 1]

29 May 2020

PONE-D-20-03354R1

An in vitro study to assess the effect of hyaluronan-based gels on muscle-derived cells: Highlighting a new perspective in regenerative medicine

PLOS ONE

Dear Dr. Schiraldi,

Thank you for submitting your manuscript to PLOS ONE. After careful consideration, we feel that it has merit but does not fully meet PLOS ONE’s publication criteria as it currently stands. Therefore, we invite you to submit a revised version of the manuscript that addresses the points raised during the review process.

The manuscript has been carefully revised and improved. There is an issue related to the statistics that should be addressed as suggested by reviewer 2.

We look forward to receiving your revised manuscript.

Kind regards,

Alberto G Passi, MD PhD

Academic Editor

PLOS ONE

Additional Editor Comments (if provided):

The manuscript has been carefully revised and improved. There is a issue related to use of statistical method T instead ANOVA. It may be addressed to confirm data robustness.

Reviewers' comments:

Reviewer's Responses to Questions

**Comments to the Author**

1. If the authors have adequately addressed your comments raised in a previous round of review and you feel that this manuscript is now acceptable for publication, you may indicate that here to bypass the “Comments to the Author” section, enter your conflict of interest statement in the “Confidential to Editor” section, and submit your "Accept" recommendation.

Reviewer #1: All comments have been addressed

Reviewer #2: (No Response)

2. Is the manuscript technically sound, and do the data support the conclusions?

Reviewer #1: (No Response)

Reviewer #2: Yes

3. Has the statistical analysis been performed appropriately and rigorously? 

Reviewer #1: (No Response)

Reviewer #2: No

4. Have the authors made all data underlying the findings in their manuscript fully available?

Reviewer #1: (No Response)

Reviewer #2: Yes

5. Is the manuscript presented in an intelligible fashion and written in standard English?

Reviewer #1: (No Response)

Reviewer #2: Yes

6. Review Comments to the Author

Reviewer #1: (No Response)

Reviewer #2: Thank you for your careful consideration of the previous review comments. There are still issues with the use of t-tests for statistical analysis. In all data sets, there are multiple levels of the independent variables, and thus multiple comparisons. Use of t-tests repeatedly compounds error and may give false positives. Thus, something like a one-way ANOVA (for normally distributed data) is required. T-tests are particularly not appropriate when multiple independent variables are evaluated (e.g., Fig. 3, where time and treatment are variables). Please address this issue.

7. PLOS authors have the option to publish the peer review history of their article (what does this mean?). If published, this will include your full peer review and any attached files.

Reviewer #1: No

Reviewer #2: No

---

## [Author Response · Author response to Decision Letter 1]

19 Jun 2020

Dear Editor,

Thank you for your consideration for our manuscript.

Hereby we submit the second revision of the manuscript number PONE-D-20-03354R1

“An in vitro study to assess the effect of hyaluronan-based gels on muscle-derived cells: Highlighting a new perspective in regenerative medicine” PLOS ONE

We considered the suggestions of reviewer 2, the manuscript was modified and corrected accordingly. 

All the minor points were addressed and modified in the text, the manuscript with track changes revisions (revised manuscript with track changes) is provided beside the “clean”version (“manuscript” without tracked changes) according to the PlosOne author guidelines.

One-way ANOVA and Tukey post hoc test were performed by JASP soft-ware (https://jasp-stats.org) and are reported in Supporting information files.

Additional Editor Comments (if provided):

The manuscript has been carefully revised and improved. There is a issue related to use of statistical method T instead ANOVA. It may be addressed to confirm data robustness.

Reviewers' comments:

Reviewer's Responses to Questions

Comments to the Author

1. If the authors have adequately addressed your comments raised in a previous round of review and you feel that this manuscript is now acceptable for publication, you may indicate that here to bypass the “Comments to the Author” section, enter your conflict of interest statement in the “Confidential to Editor” section, and submit your "Accept" recommendation.

Reviewer #1: All comments have been addressed

Reviewer #2: (No Response)

2. Is the manuscript technically sound, and do the data support the conclusions?

Reviewer #1: (No Response)

Reviewer #2: Yes

3. Has the statistical analysis been performed appropriately and rigorously?

Reviewer #1: (No Response)

Reviewer #2: No

4. Have the authors made all data underlying the findings in their manuscript fully available?

Reviewer #1: (No Response)

Reviewer #2: Yes

5. Is the manuscript presented in an intelligible fashion and written in standard English?

Reviewer #1: (No Response)

Reviewer #2: Yes

6. Review Comments to the Author

Reviewer #1: (No Response)

Reviewer #2: Thank you for your careful consideration of the previous review comments. There are still issues with the use of t-tests for statistical analysis. In all data sets, there are multiple levels of the independent variables, and thus multiple comparisons. Use of t-tests repeatedly compounds error and may give false positives. Thus, something like a one-way ANOVA (for normally distributed data) is required. T-tests are particularly not appropriate when multiple independent variables are evaluated (e.g., Fig. 3, where time and treatment are variables). Please address this issue.

We have included for Fig 3, Fig 5A and Fig 6A, the one-way ANOVA and Tukey post hoc test by JASP soft-ware (https://jasp-stats.org). In fact in this way we efficiently compared the different treatments and data sets. On the figures we inserted the symbols corresponding to p values for the diverse comparisons. All the data are reported as tables of the software outcomes and included as Supporting information.

Please find following the specific response to each point raised by reviewer and inserted in the manuscript accordingly:

Line 247-250 page 11 in the statistical analyses paragraph.

Line 298, 303, 305, 308 page 13 in the MMT-test results referring to Fig3.

Line 329-331 page 14 in the fig3 legend.

Line 366,368,371 page 15 for qRT-PCR results and 386-393 for fig5A legend.

Line 408 page 17 qRT-PCR results and 413-421 page 17 for fig6A legend.

We think that this further in depth analyses of experimental data set is improving the manuscript quality and for this reason we wish to thank the reviewer for the suggestion.

7. PLOS authors have the option to publish the peer review history of their article (what does this mean?). If published, this will include your full peer review and any attached files.

Do you want your identity to be public for this peer review? For information about this choice, including consent withdrawal, please see our Privacy Policy.

Reviewer #1: No

Reviewer #2: No

---

## [Decision Letter · Decision Letter 2]

1 Jul 2020

An in vitro study to assess the effect of hyaluronan-based gels on muscle-derived cells: Highlighting a new perspective in regenerative medicine

PONE-D-20-03354R2

Dear Dr. Schiraldi,

We’re pleased to inform you that your manuscript has been judged scientifically suitable for publication and will be formally accepted for publication once it meets all outstanding technical requirements.

Kind regards,

Alberto G Passi, MD PhD

Academic Editor

PLOS ONE

Additional Editor Comments (optional):

In this revised version of the manuscript the Authors properly addressed all concerns raised by the reviewers.

Reviewers' comments:

Reviewer's Responses to Questions

**Comments to the Author**

1. If the authors have adequately addressed your comments raised in a previous round of review and you feel that this manuscript is now acceptable for publication, you may indicate that here to bypass the “Comments to the Author” section, enter your conflict of interest statement in the “Confidential to Editor” section, and submit your "Accept" recommendation.

Reviewer #2: All comments have been addressed

2. Is the manuscript technically sound, and do the data support the conclusions?

Reviewer #2: Yes

3. Has the statistical analysis been performed appropriately and rigorously? 

Reviewer #2: Yes

4. Have the authors made all data underlying the findings in their manuscript fully available?

Reviewer #2: Yes

5. Is the manuscript presented in an intelligible fashion and written in standard English?

Reviewer #2: Yes

6. Review Comments to the Author

Reviewer #2: (No Response)

7. PLOS authors have the option to publish the peer review history of their article (what does this mean?). If published, this will include your full peer review and any attached files.

Reviewer #2: **Yes: **Stephanie Seidlits

---

## [Editor Report · Acceptance letter]

13 Jul 2020

PONE-D-20-03354R2 

An in vitro study to assess the effect of hyaluronan-based gels on muscle-derived cells: Highlighting a new perspective in regenerative medicine 

Dear Dr. Schiraldi:

I'm pleased to inform you that your manuscript has been deemed suitable for publication in PLOS ONE. Congratulations! Your manuscript is now with our production department. 

Kind regards, 

on behalf of

Prof. Alberto G Passi 

Academic Editor

PLOS ONE